# Genomic screening of 16 UK native bat species through conservationist networks uncovers coronaviruses with zoonotic potential

Cedric C. S. Tan [1,2,11], Jahcub Trew[3,11], Thomas P. Peacock [4,11], Kai Yi Mok [4], Charlie Hart [3], Kelvin Lau [5], Dongchun Ni [6], C. David L. Orme [3], Emma Ransome[3], William D. Pearse[3], Christopher M. Coleman[7], Dalan Bailey [8], Nazia Thakur [8,9], Jessica L. Quantrill[4], Ksenia Sukhova [4], Damien Richard[1], Laura Kahane[3], Guy Woodward [3], Thomas Bell [3], Lisa Worledge [10], Joe Nunez-Mino[10], Wendy Barclay [4], Lucy van Dorp [1], Francois Balloux [1] & Vincent Savolainen [3] ✉

There has been limited characterisation of bat-borne coronaviruses in Europe. Here, we screened for coronaviruses in 48 faecal samples from 16 of the 17 bat species breeding in the UK, collected through a bat rehabilitation and conservationist network. We recovered nine complete genomes, including two novel coronavirus species, across six bat species: four alphacoronaviruses, a MERS-related betacoronavirus, and four closely related sarbecoviruses. We demonstrate that at least one of these sarbecoviruses can bind and use the human ACE2 receptor for infecting human cells, albeit suboptimally. Additionally, the spike proteins of these sarbecoviruses possess an R-A-K-Q motif, which lies only one nucleotide mutation away from a furin cleavage site (FCS) that enhances infectivity in other coronaviruses, including SARS-CoV-2. However, mutating this motif to an FCS does not enable spike cleavage. Overall, while UK sarbecoviruses would require further molecular adaptations to infect humans, their zoonotic risk warrants closer surveillance.

The majority of emerging infectious diseases in humans are zoonotic—arising from the animal-to-human transmission of a pathogen[1]—and more than 70% originate in wildlife[2].

*Coronaviridae* is a diverse family of viruses that can infect a broad range of animals and are prone to zoonotic spillovers. There are seven major coronaviruses known to infect humans: SARS-CoV-2 is the agent of the COVID-19 pandemic whose direct ancestor has not yet been identified, but its closest relatives have been isolated from horseshoe bats. SARS-CoV caused a major international outbreak in 2002–2004 with around 8000 recorded cases and at least 774 deaths[3]. MERS-CoV fuels recurrent disease outbreaks in humans through repeated host jumps from its reservoir in camels[4]. Four coronaviruses (HCoV-229, -NL63, -OC43 and -HKU1) circulate endemically in humans and their ancestral reservoirs are believed to be species of bats and rodents, with host jumps into humans likely facilitated by other mammals as bridging hosts[5–7]. In addition, multiple host jumps from animals into humans leading to isolated or small clusters of cases have been documented for other coronavirus species[5]. Given the current health burden exerted by coronaviruses and the risk they pose as possible agents of future epidemics and pandemics, surveillance of animal-borne coronaviruses should be a public health priority. Indeed, the discovery and characterisation of the diversity of coronaviruses

harboured by mammals across the world is the first step for designing pre-emptive measures to minimise human or animal exposure. Here, we focus on bats since some human coronaviruses have their ancestral origins in some of these host species.

Several studies over the last decade have screened bats across Asia, Africa, the Middle East and Europe for coronaviruses, finding anywhere from 1.5-23% coronavirus prevalence in animals tested[8-19]. A selection of studies representing the diversity of previous screening efforts is listed in Supplementary Data 1. These prevalence estimates were primarily obtained via a reverse transcription PCR (RT-PCR) using degenerate primers designed to target most coronavirus species (i.e., pan-coronavirus primers; Supplementary Data 1). Given the vast diversity of coronaviruses, including those yet to be discovered, it is difficult to design primers that can amplify and capture the full diversity of coronaviruses. Our own comparative analysis of published primer sets show that existing RT-PCR assays[20-24] may underestimate coronavirus prevalence (Supplementary Fig. 1). Difficulties with primer design is exacerbated by low RNA concentrations in field samples and RNA degradation, so the large variability in prevalence estimates in these studies may be due to the sensitivity of the primer set used rather than the epidemiology of bat coronaviruses. While sample RNA quality remains mainly dependent on sample collection and laboratory practices, because untargeted RNA sequencing does not require a priori knowledge of sequence information, it provides a more accurate estimate of viral diversity and prevalence. Hence, we chose this approach over RT-PCR specifically to screen UK bats for coronaviruses.

For a zoonotic event to occur, a virus must be able to transmit efficiently between animals and humans and be able to infect and replicate in human cells through interactions with host cellular machinery. Additionally, zoonotic pathogens of most concern are those that can transmit efficiently between humans. As such, the true zoonotic potential (i.e., the likelihood of infecting humans in the future) of a virus can only be determined by assessing all these prerequisite abilities. Therefore, excluding unethical experimental infection or transmission studies involving humans, assessments of zoonotic potential should be done via alternative approaches. These include in silico analyses that determine the degree of sequence and structural homology to other known and closely related human-infecting viruses[11,25]. Even more compelling evidence for zoonotic potential can be obtained in vitro, and one of the most direct assessments is to isolate and test the infectivity of novel viruses in human cells[25,26]. However, this would increase the risk of exposure to these potentially infectious agents, necessitating stringent biosafety precautions. In addition, isolation of novel viruses via cell culture, without prior knowledge of their cell tropism and receptor usage, can be challenging.

A low-risk and effective alternative is to measure the binding efficiency of viral entry proteins to host receptors[10], or to assess efficiency of viral entry into human-cell lines via a pseudovirus assay[25], which expresses only the viral entry protein in a non-infectious reporter system. While observed binding and cellular entry in these low-risk assays do not show that a virus can replicate effectively in human cells, infect humans, or transmit efficiently between humans, they provide an indication of which human-cell receptors can be exploited by novel viruses, which are one key determinant of viral infection. However, despite the importance of functional validation, many studies to date fall short of providing in vitro or even in silico assessments of zoonotic risk (Supplementary Data 1).

There are 17 bat species that breed in the United Kingdom (UK), most of which often roost in domestic buildings, churches, barns and other man-made structures. This frequent habitat overlap with humans places bats in close proximity to domesticated and farmed animals, which can serve as potential bridging hosts for transmitting bat-borne viruses to humans[27]. However, multiple factors have to align for the successful emergence of a zoonotic pathogen in humans, including the frequency of exposure, the ability of the pathogen to infect humans and its capacity for onward human-to-human transmission[28]. The relative risks of these various factors for zoonotic spillover remains largely unknown and may vary depending on pathogen and geographical context. All UK bat species are protected by law across the UK with licences required for work related to bats. Therefore, although direct contact is rare among the general public, it is far more common for the small proportion of the population comprising bat scientists, ecologists, conservationists and bat rehabilitators that undertake regular research, monitoring, surveillance, and bat rehabilitation work.

Only two studies screening UK bats for coronaviruses have been conducted to date[11,19]. The first, published over a decade ago, screened seven bat species and detected alphacoronaviruses in Daubenton's bat and Natterer's bat (*Myotis daubentonii* and *M. nattereri*, respectively)[19]. The other, from 2021, screened faecal samples from lesser horseshoe bats (*Rhinolophus hipposideros*) and recovered the whole genome sequence of a single sarbecovirus, RhGB01 (MW719567)[11]. However, neither study provided direct in vitro assessments of zoonotic risk. Accordingly, the viral diversity and zoonotic potential of UK bat viruses remains largely unknown. This is equally true of viruses in most other UK mammals. However, given that the evolutionary origins of many coronaviruses of human health concern can be traced back to bats, assessing their zoonotic potential in UK bats is a top priority, before moving on to other animal groups.

To address this knowledge gap, we used an existing UK network of bat rehabilitators and conservationists to collect faecal samples from UK bats. Faeces from all but one bat species breeding in the UK (the grey long-eared bat, *Plecotus austriacus*, the rarest species in the UK) were collected and subsequently screened using deep RNA sequencing to characterise the genomic diversity of bat-borne coronaviruses in the UK. To assess their zoonotic potential, we then tested the ability of a subset of these coronaviruses to bind human-cell receptors in vitro, which is a key requisite for human infection.

## Results

### Untargeted RNA sequencing recovers nine complete coronavirus genomes, including two new species

We performed deep RNA sequencing on 48 faecal samples from 16 of the 17 UK breeding bat species, with wide geographic coverage and spanning 2 years of sampling (Supplementary Fig. 2). Through taxonomic assignment of sequencing reads using Kraken2[29], we detected the presence of at least 30 viral families, 53% of which primarily infect non-mammalian hosts such as plants, insects and bacteria (Supplementary Fig. 3a). In addition, the total relative abundance of viral species that infect non-mammalian hosts was significantly higher than that for mammalian viruses (two-sided Mann–Whitney $U$ test, $U = 1393$, $P = 0.004$; Supplementary Fig. 3b). These findings indicate that the faecal 'virome' in UK bats largely comprises viruses that do not necessarily infect them, nor other mammals, including humans.

We next focused on coronaviruses due to their relevance to human health and recovered nine complete genomes (96-100% completeness; assessed by CheckV[30]) and five partial contigs (<3%) across six UK bat species (*M. daubentonii, Pipistrellus pipistrellus, P. pygmaeus, P. auritus, R. ferrumequinum,* and *R. hipposideros*), detecting coronaviruses amongst 29% of the samples. The nine complete genomes were assessed by CheckV to be of high quality[30] and read alignments to these genomes indicated an even coverage of reads with a median coverage of 548–7958 reads per position (Supplementary Fig. 4 and Table 1).

A global phylogenetic tree based on alignment-free genetic distances[31] revealed the genus and subgenus membership of these new coronaviruses (Fig. 1a; see "Methods"). We then followed with local maximum-likelihood phylogenetic analyses to determine their precise placement within each subgenus (Fig. 1b–d). These phylogenetic

analyses reveal that the nine novel genomes we recovered comprise four alphacoronaviruses from the *Pedacovirus* subgenus, five beta-coronaviruses including one merbecovirus, and four sarbecoviruses (Fig. 1). Three of the coronaviruses recovered from *M. daubentonii* (which we call MdGB01-03) form a well-supported clade with other pedacoviruses isolated from the same bat species in Denmark (Fig. 1b). One coronavirus sequenced from *P. pipistrellus* (PpiGB01) falls as a sister lineage to the above clade. Another coronavirus from *P. auritus* (PaGB01) is related to MERS-CoV-like merbecoviruses isolated from *Hypsugo*, *Pipistrellus*, and *Vespertilio* spp. from Western Europe and China (Fig. 1c). Four coronaviruses isolated from *R. ferrumequinum* and *R. hipposideros* (RfGB01-02 and RhGB07-08, respectively) are closely related to the previously described UK bat sarbecovirus, RhGB01[11] (Fig. 1d).

Of the nine coronaviral genomes recovered here, two represent new species. Indeed, pedacovirus PpiGB01 from *P. pipistrellus* was relatively divergent from its closest match, a pedacovirus previously isolated from *M. daubentonii* (less than 81% nucleotide sequence identity; Table 1). Similarly, merbecovirus PaGB01 shares less than 82% sequence identity to its closest match, a merbecovirus isolated from *P. kuhlii* in Italy (Table 1). Overall, our coronavirus screening efforts have extended our knowledge of the existing diversity of coronaviruses. Further, looking at their genomic structures, we identified one new gene in each of these new species (Supplementary Fig. 5).

Viruses that are able to infect a broad range of hosts have been associated with a higher risk of emerging as infectious diseases among humans[32,33]. Here, the four sarbecovirus genomes, representing one viral species, were recovered from two distinct horseshoe bat species, *R. ferrumequinum* and *R. hipposideros*. RhGB07, RhGB08, RfGB01, RfGB02 share 97–100% identity with RhGB01 previously described in *R. hipposideros*[11]. To better understand how these viruses might be shared among the two hosts, we compared the habitat distributions of both horseshoe bat species. The two horseshoe bat species share a large proportion of their habitats, with 33% of their occurrence records reported at the same geographical coordinates. Furthermore, species distribution modelling predicted that 45% of the total land area that may form suitable habitats for the two species is shared (Supplementary Fig. 6a). Since the two *Rhinolophus* species can share roosts[34], these results indicate a potentially high frequency of direct contact, which may facilitate viral sharing and thus account for the isolation of RhGB01-like sarbecoviruses that are closely related from these two species.

To extend this analysis, we examined both observed and predicted distributions of all 17 UK breeding bat species to identify potential viral sharing hotspots for future surveillance work. By analysing 42,953 occurrence records, we identified three regions near Bristol, Birmingham and Brighton with particularly high species diversity (the most diverse regions having 16 species in a single 5 × 5 km grid; Supplementary Fig. 6b). In addition, we identified regions within the UK, especially in Wales and the South coast of England where the habitats of the greatest number of different bat species are predicted to coincide (Supplementary Fig. 6c). Alongside an understanding of the ecology of native species, including co-roosting and foraging behaviours, such information is a useful resource for future surveillance studies, and for prioritising focal areas of zoonotic risk.

## Sarbecoviruses recovered from UK bats can bind the human ACE2 receptor for cellular entry

We tested whether representatives of the newly identified UK coronaviruses (the sarbecoviruses RhGB07 and RfGB02, the merbecovirus PaGB01, and the outlier pedacovirus PpiGB01) could use human cellular receptors for viral entry as a proxy for assessing their zoonotic potential. We successfully incorporated the spike proteins of these UK coronaviruses into lentivirus-based pseudoviruses (see "Methods" and Supplementary Fig. 7c). We then tested the ability of these spike-expressing pseudoviruses to infect human cells expressing the human

**Table 1 | Summary statistics for novel coronavirus genomes assembled in this study**

| Host species | Common name | Genome name | Subgenus | Length (bp) | Closest hit Accession | Closest hit Name | BLASTn identity (%) | Prop. of query aligned (%) | CheckV completeness | Median coverage |
|---|---|---|---|---|---|---|---|---|---|---|
| *Rhinolophus ferrumequinum* | Greater horseshoe bat | RfGB01 | *Sarbecovirus* | 29,308 | MW719567 | RhGB01 | 98.1 | 99.7 | 97 | 689 |
| | | RfGB02 | *Sarbecovirus* | 29,375 | MW719567 | RhGB01 | 98.1 | 99.4 | 97 | 7178 |
| *Rhinolophus hipposideros* | Lesser horseshoe bat | RhGB07 | *Sarbecovirus* | 29,224 | MW719567 | RhGB01 | 97.9 | 100 | 97 | 3809 |
| | | RhGB08 | *Sarbecovirus* | 29,217 | MW719567 | RhGB01 | 98 | 100 | 96 | 548 |
| *Plecotus auritus* | Brown long-eared bat | PaGB01 | *Merbecovirus* | 30,018 | MG596803 | *P. kuhlii* MERS-related CoV | 81.5 | 99.7 | 99 | 6237.5 |
| *Pipistrellus pipistrellus* | Common pipistrelle | PpiGB01 | *Pedacovirus* | 28,247 | MN535731 | *M. daubentonii* pedacovirus | 80.8 | 82.9 | 100 | 7438 |
| *Myotis daubentonii* | Daubenton's bat | MdGB01 | *Pedacovirus* | 28,224 | MN535731 | *M. daubentonii* pedacovirus | 95.5 | 99.6 | 100 | 7938 |
| | | MdGB02 | *Pedacovirus* | 28,010 | MN535733 | *M. daubentonii* pedacovirus | 95.4 | 99.8 | 100 | 7874 |
| | | MdGB03 | *Pedacovirus* | 28,227 | MN535733 | *M. daubentonii* pedacovirus | 95.4 | 99.8 | 100 | 7958 |

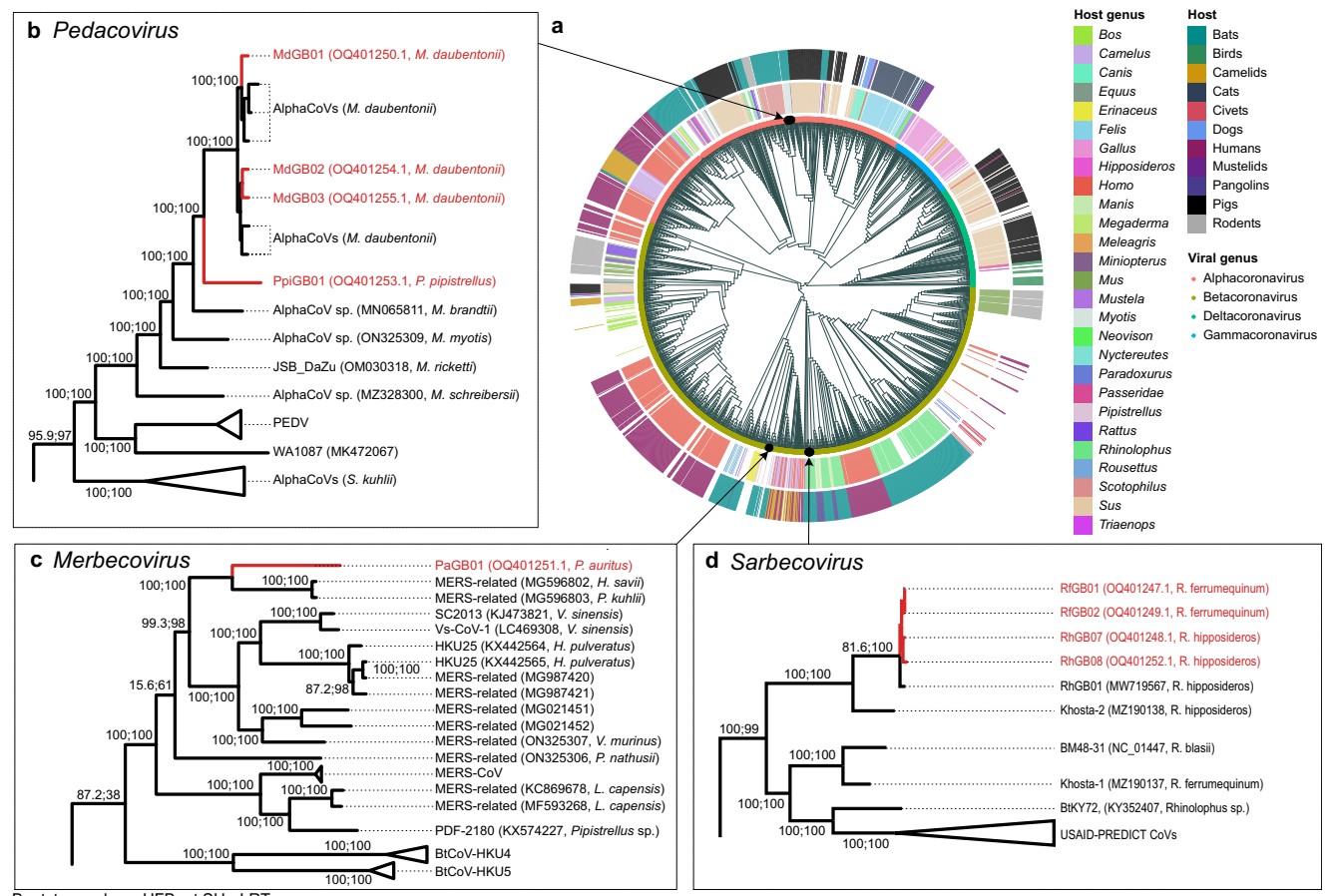

**Fig. 1 | Phylogenetic placement of novel coronaviruses. a** Alignment-free phylogeny of the global diversity of coronavirus genomes ($n = 2118$) and our nine novel genomes. Host genus (inner ring) and their broader host groups (outer ring) are annotated. Local maximum-likelihood trees of **b** pedacoviruses ($n = 106$), **c** merbecoviruses ($n = 113$) and **d** sarbecoviruses ($n = 534$).

receptors, angiotensin-converting enzyme 2 (hACE2), dipeptidyl peptidase-4 (hDPP4) and aminopeptidase N (hAPN), which are the primary receptors exploited by SARS-CoV-2, MERS-CoV and HCoV-229E, respectively.

None of the spike pseudoviruses of the UK coronaviruses could enter cells using any of the receptors except RhGB07, which showed significantly higher entry into cells overexpressing hACE2 compared to those not expressing hACE2 (Fig. 2a; $P < 0.0001$). As expected, SARS-CoV-2, MERS-CoV and HCoV-229E showed significantly higher entry into cells overexpressing hACE2, hDPP4 and hAPN, respectively (Fig. 2a; $P < 0.0001$). Further, VSV-G-expressing pseudovirus, which enters cells using ubiquitous protein receptors, showed comparably high entry across all groups (Fig. 2a). In addition, using biolayer-inteferometry (BLI), we confirmed that the RhGB07 spike is able to bind hACE2 with a dissociation constant, $K_d = 253$ nM (Fig. 2b). However, the binding affinity of RhGB07 spike to hACE2 is ~17-fold lower than that for the SARS-CoV-2 spike ($K_d = 15$ nM) (Fig. 2b).

Given the lower binding affinity of RhGB07 spike compared to SARS-CoV-2, we then investigated if, like SARS-CoV-2, RhGB07 spike-expressing pseudoviruses can infect human cells expressing lower (HEK293T-hACE2—HEK293Ts stably transduced with hACE2; Supplementary Fig. 7a) or physiological levels of hACE2 (Calu-3 lung, and Caco-2 colorectal cell lines). Alongside this, we tested the entry of RfGB02, PaGB01 and PpiGB01 spike pseudoviruses in the same cell lines in case they use a human receptor not otherwise tested as in Fig. 2a. As positive controls, we included the spike proteins from other coronaviruses, BANAL-20-52/SARS-CoV-2 (wild-type Wuhan-Hu-1 with D614G), MERS-CoV, and HCoV-229E, which can efficiently enter these cell lines using hACE2[8,35], hDPP4 and hAPN, respectively. We also included the negative control, RaTG13, which can bind hACE2 but cannot enter cells expressing lower or physiological levels of hACE2[35]. As expected, RaTG13 could not enter any of these cell lines, while all positive controls showed significantly higher entry into these cell lines than "bald" pseudoviruses not expressing any spike protein ($P < 0.01$; Fig. 2c). In contrast, none of the UK spike pseudoviruses tested, including RhGB07, displayed significant entry into any of these human-cell lines ($P > 0.05$; Fig. 2c).

Separately, we asked if other host proteins are necessary for efficient cellular entry of the UK coronaviruses. In particular, the transmembrane serine protease 2 (TMPRSS2) protease has been shown to greatly enhance the entry efficiency of MERS-CoV[36] and HCoV-229E[37] spike pseudoviruses into human cells. However, PaGB01 and PpiGB01, which fall in the same subgenus as MERS-CoV and HCoV-229E, respectively, could not enter TMPRSS2-overexpressing cells (Fig. 2d).

RhGB07 can bind and use overexpressed hACE2 for cellular entry but RfGB02 cannot, despite the high 98% sequence identity of their spike proteins. This begs the questions as to how RhGB07 might have acquired the ability to use hACE2, and whether this might be associated with the usage of bat ACE2 orthologues. To investigate this, we tested the entry of RhGB07 and RfGB02 spike pseudoviruses into human cells expressing the ACE2 orthologues from four bat species (*R. ferrumequinum*, *R. pusillus*, *Myotis lucifugus*, and *Rousettus leschenaultia*). These bat ACE2 proteins all expressed robustly, albeit to slightly differing efficiencies (Supplementary Fig. 7b), as seen previously[38]. Unfortunately, throughout the course of this study, there was no publicly available ACE2 sequence for *R. hipposideros* (from

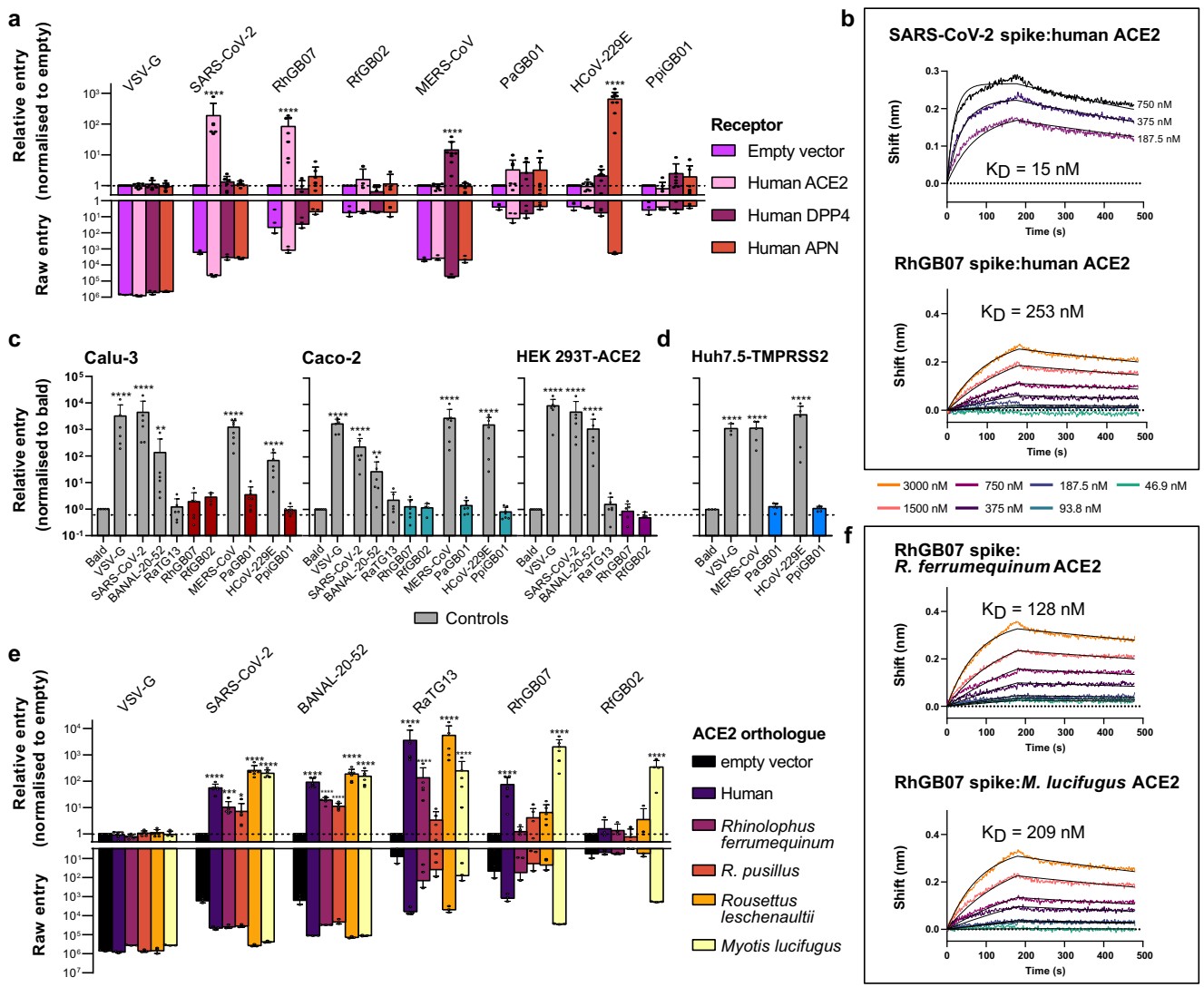

**Fig. 2 | RhGB07 can bind and use human ACE2 for cell entry in vitro. a** Entry of different spike pseudoviruses expressing viral glycoproteins into HEK293T cells transfected with (**a**) human receptors known to allow entry of human coronaviruses or (**e**) ACE2 homologues from different species. **a, e** The raw entry values for each pseudoviruses were normalised by their entry into cells transfected with a vector containing no receptor sequence (i.e., "empty"). The raw entry values of representative repeats (*n* = 3 independent experiments) are also shown for direct comparisons of absolute entry. Representative biolayer interferometry binding curves showing the association and dissociation of SARS-CoV-2 and RhGB07 spike proteins with (**b**) hACE2 (*n* = 2 and 3 independent experiments, at three and seven protein concentrations, respectively), or (**f**) with *R. ferrumequinum* or *M. lucifugus* ACE2 (*n* = 1 independent experiment, at seven protein concentrations). **c** Entry of pseudoviruses into different "normal" human-cell lines that stably express lower or physiological levels of hACE2. All entry measurements are normalised to those for the "bald" pseudovirus not expressing any spike protein. **d** Entry of pseudoviruses

into Huh7.5 cells transduced with a human TMPRSS2 vector, normalised to "bald". Data from panels (**a**, **c**–**e**) are compiled from *n* = 3–8 independent experiments and plotted as mean + s.d. Statistical significance was determined by (**a**, **e**) two-way ANOVA or **c**, **d** one-way ANOVA on log-transformed data (after determining log normality by the Shapiro–Wilk test and QQ plot) with multiple comparisons against "empty" vector or "bald" pseudovirus, respectively. *0.05 ≥ *P* > 0.01; **0.01 ≥ *P* > 0.001; ***0.001 ≥ *P* > 0.0001; ****P ≤ 0.0001. Exact *P* values annotated for each graph are as follows (from left to right), (**a**) <0.0001, <0.0001, <0.0001, <0.0001; (**c**, Calu-3) < 0.0001, <0.0001, 0.001, <0.0001, <0.0001; (**c**, Caco-2) < 0.0001, <0.0001, 0.022, <0.0001, <0.0001; (**c**, HEK293T-ACE2) < 0.0001, <0.0001, <0.0001; **d** <0.0001, <0.0001, <0.0001, <0.0001. (**e**, SARS-CoV-2) < 0.0001, 0.0003, 0.12, <0.0001, <0.0001 (**e**, BANAL-20-52) < 0.0001, <0.0001, <0.0001, <0.0001, <0.0001; (**e**, RatG13) < 0.0001, <0.0001, <0.0001, <0.0001 (**e**, RhGB07) < 0.0001, <0.0001; (**e**, RfGB02) < 0.0001.

which RhGB07 was recovered), and no ACE2 transcripts could be identified directly from our metatranscriptomic libraries.

Nevertheless, we detected significant cell entry but only through *M. lucifugus* ACE2 (*P* < 0.0001; Fig. 2e), and neither RhGB07 nor RfGB02 could use *R. ferrumequinum* ACE2 receptors, despite RfGB02 being sampled from this species. In contrast, SARS-CoV-2, BANAL-20-52 and RaTG13 were all able to efficiently use *R. ferrumequinum* ACE2 (*P* < 0.0001; Fig. 2e), indicating that this ACE2 construct could allow sarbecovirus entry. Surprisingly, BLI measurements indicate detectable binding of RhGB07 spike to both *R. ferrumequinum* and *M. lucifugus* ACE2 (Fig. 2f), which means that RhGB07 can bind *R.*

*ferrumequinum* ACE2 but not efficiently enter cells expressing this receptor. This highlights that binding of host ACE2 alone may not be sufficient for efficient viral entry, and that other host cell–virus interactions (e.g., presence of suitable co-receptors) may be required. Taken together, our results suggest that RhGB01-like sarbecoviruses may not be using ACE2 to infect their native *Rhinolophus* hosts.

### Structural and sequence features of RhGB07 spike explain detectable but inefficient usage of hACE2

To gain deeper insight into the results of the above assays, we used the AlphaFold2 artificial intelligence model[39] to predict the 3D structure of

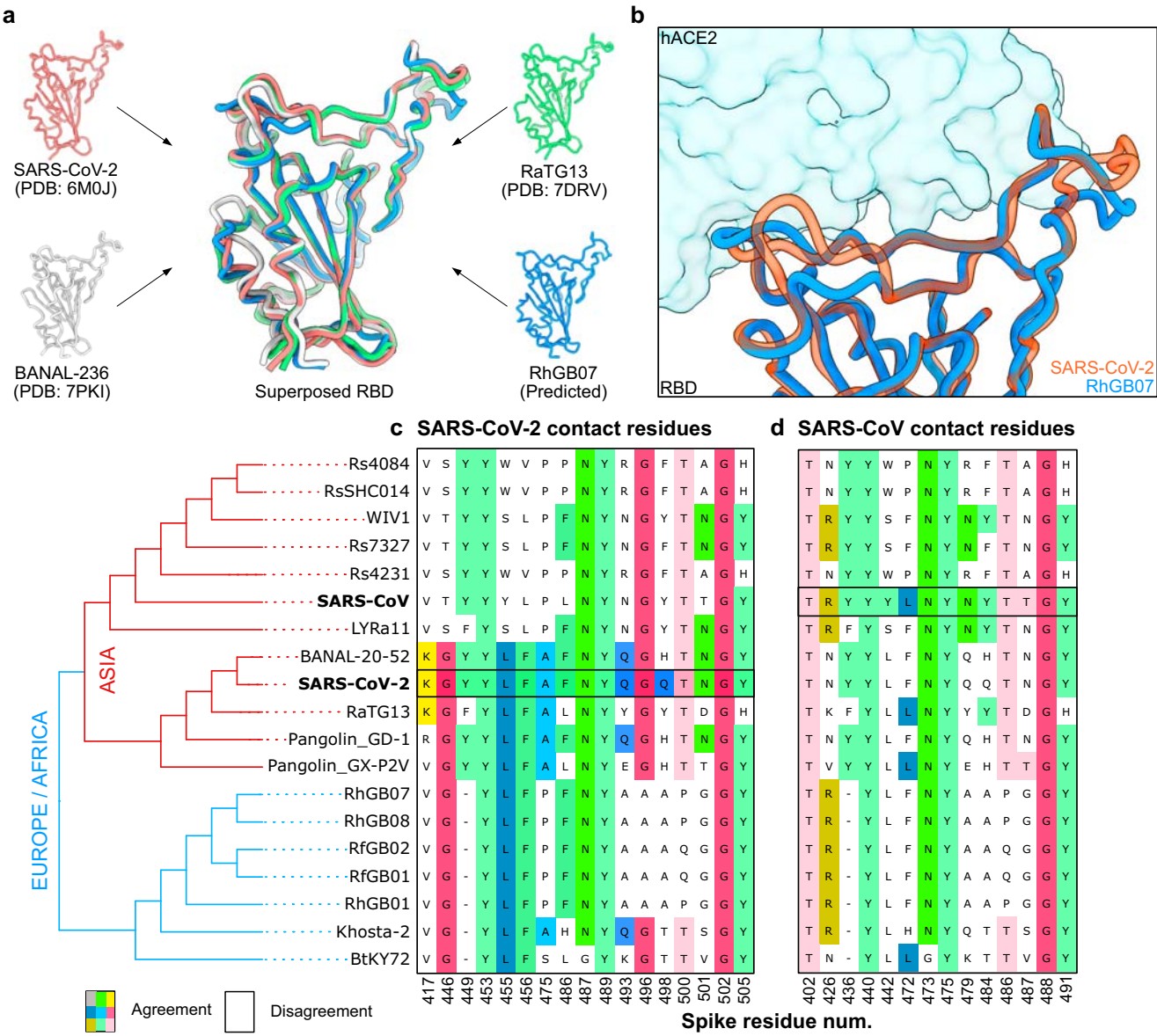

**Fig. 3 | Structural and sequence features of RhGB01-like sarbecoviruses. a** The solved RBD structures of SARS-CoV-2, RaTG13, BANAL-236 (close relative of BANAL-20-52[8]) and the AlphaFold2-predicted structure of RhGB07 were superposed. **b** The 3D surfaces of the RBD-hACE2 binding interface for SARS-CoV-2 and RhGB07. Alignment of sarbecovirus spike proteins showing the conservation of key contact residues involved interactions between (**c**) SARS-CoV spike and (**d**) SARS-CoV-2 spike with hACE2. The sequences shown in the alignments are from Asian, European and African sarbecoviruses that have been shown to bind hACE2[42–44]. These sequences were ordered based on their genetic relatedness, as inferred from a consensus maximum-likelihood phylogenetic tree reconstructed from their whole genomes (bottom left).

the receptor-binding domain (RBD) of the RhGB07 spike protein. We then compared it to the resolved RBD structures of SARS-CoV-2[40], BANAL-236 (a close relative of BANAL-20-52)[8], and RaTG13[35] bound to hACE2. Superposition of the RBD structures showed high structural conservation across all four sarbecoviruses (Fig. 3a). In addition, the 3D structure of the RhGB07 RBD near the RBD-hACE2 binding interface was highly similar to that of SARS-CoV-2 (Fig. 3b), which was confirmed by comparing the area of contact surface (894 Å² and 850 Å², respectively; Supplementary Fig. 8).

These findings account for the ability of the RhGB07 spike protein to bind hACE2 (Fig. 2a, b). To understand why RhGB07 spike pseudoviruses could not enter cells expressing the ACE2 receptor at physiological levels (Fig. 2c), we compared the level of conservation at key RBD residues of SARS-CoV-2[40] and SARS-CoV[41] in contact with hACE2. This included sarbecoviruses isolated in Asia, Europe, and Africa, which bind hACE2 with various affinities[42–44]. All these sarbecoviruses

showed conservation at more than half of SARS-CoV-2 (Fig. 3c) or SARS-CoV (Fig. 3d) contact residues, with high levels of conservation at certain positions like Y453, N487, Y489, G502 and Y505 (SARS-CoV-2 numbering; Fig. 3c). Previous deep mutational scanning experiments showed that all these positions, except Y453, cannot be mutated without considerable loss of hACE2 binding affinity[45], indicating that contact residues are important determinants of hACE2 binding. Notably, the novel RhGB01-like sarbecoviruses share only 9/17 SARS-CoV-2 (Fig. 3c) and 7/14 SARS-CoV contact residues (Fig. 3d). This is slightly below RaTG13, which shares 11/17 SARS-CoV-2 and 8/14 SARS-CoV contact residues, respectively. In contrast, BANAL-20-52 shares 15/17 contact residues with SARS-CoV-2.

These results indicate poorer conservation of these key contact residues in UK sarbecoviruses, which would explain the relatively lower hACE2 usage efficiency, and hence the ability to infect human cells, of RhGB07 and RaTG13 compared to BANAL-20-52. Similarly, the

poorer conservation of SARS-CoV-2 contact residues in BtKY72 than Khosta-2 (Fig. 3c) may explain the lower binding affinity to hACE2 of the former[43]. Notably, RhGB07–but not RfGB02–could enter cells using hACE2 (Fig. 2a), despite their spike proteins sharing the same variants at all SARS-CoV-2 key contact residues (Fig. 3c). The RfGB02 spike has 26 amino acid mutations relative to RhGB07, and only four of these were within the RBD (K337N, H432L, T470A, P487Q; Supplementary Data 2). As such, either the residues at these positions are, in addition to the SARS-CoV-2 contact residues, important mediators of hACE2 binding, or the remaining 22 non-RBD mutations have caused structural changes that reduce the binding affinity to hACE2, or both. Further experiments delineating these mutational effects could help to shed light on the molecular determinants of sarbecoviral entry into human cells.

Remarkably, the RhGB01-like sarbecoviruses all possess a R-A-K-Q sequence at the S1/S2 cleavage site (spike residues 669–672; Supplementary Fig. 9a), which is one nucleotide away (Gln/C**A**A to Arg/C**G**A) from the canonical R-X-K/R-R motif, a potential furin cleavage site (FCS) that allows cleavage by host furin-like proteases, enhancing the ability of many coronaviruses, including SARS-CoV-2, to infect human cells[46,47]. This R-A-K-Q motif is also found in Khosta-2[48], a sarbecovirus recovered from *R. hipposideros* in Khosta, Russia, which is at the southeastern extremes of Europe, but not in BtKY72 from *Rhinolophus* sp. in Kenya[49] or other sarbecoviruses isolated from Asia. However, Western blot analyses indicated that even when we mutated R-A-K-Q to R-A-K-R (i.e., a Q672R mutation), the RhGB07 spike is not cleaved by endogenous human host proteases during trafficking to the cell surface (Supplementary Fig. 9b). Previous studies have shown that the FCS on SARS-CoV-2 spike (681-RRAR-684) lies on an extended flexible loop that protrudes out of the spike structure, which allows access by host furin[50,51]. Also, it has been shown that deletions that shortened this extended loop, while leaving the multi-basic site intact, prevented efficient cleavage of SARS-CoV-2 spike, which was likely due to reduced accessibility of the FCS[47]. This loop is seven residues shorter in RhGB01-like sarbecoviruses than in SARS-CoV-2 (Supplementary Fig. 9b), which may explain why no cleavage was observed for the RhGB07 R-A-K-R pseudovirus mutant.

Finally, a recombination analysis of the RhGB01-like and other representative sarbecoviruses indicates a high prevalence of recombination (Supplementary Information; Supplementary Fig. 10), which may accelerate adaptation for infecting novel hosts. Given these findings, the current zoonotic risk of sarbecoviruses in UK bats, while small, cannot be ignored and warrants more extensive surveillance of bats at the national scale.

## Discussion

The emergence of the COVID-19 pandemic in 2019 is a sobering reminder of the massive impact of zoonotic viruses on global health and economy. Despite this, genomic surveillance in wildlife remains limited. In this study, we used an existing network of bat rehabilitators and conservationists to obtain geographically and temporally diverse samples from almost all bat species in the UK. We argue that this can be a sustainable and effective surveillance model to identify and characterise novel animal-borne viruses that may or may not yet be able to infect humans but might evolve the ability to do so in the future.

We provided evidence that at least one sarbecovirus isolated from UK horseshoe bats can bind hACE2 in vitro and discuss these patterns relative to our in silico analyses. Crook et al.[11] performed a contact residue analysis, similar to the one we report in Fig. 3c, d, on RhGB01 and suggested that moderate similarity in its key contact residues indicates that it is unlikely to bind hACE2. However, our in vitro (Fig. 2) and in silico (Fig. 3) results highlight that despite having only moderate conservation of key contact residues, RhGB07 can bind and use hACE2. Additionally, the spike of RhGB07, but not RfGB02, can use hACE2 for cellular entry, despite identical conservation levels to SARS-CoV-2 at

key contact residues. These findings indicate that assessing the conservation of key contact residues (Fig. 3c, d) may have limited predictive power for whether a spike protein can bind hACE2, possibly due to multiple structural configurations allowing hACE2 binding. This is further evidenced by the different contact residues for SARS-CoV and SARS-CoV-2.

Our findings indicate that the RhGB01-like sarbecoviruses likely require further adaptations, particularly in their spike proteins, before they can make a zoonotic jump. Notably, single mutations of some of the SARS-CoV-2 contact residues in sarbecoviral spike proteins have been shown to enable binding of ACE2 from novel host species and improve binding affinity by greater than fivefold[43]. In addition, a single T403R mutation in the RaTG13 spike has been shown to allow the virus to infect human cells[52]. Given this, we speculate that the genetic barrier precluding effective hACE2 usage for cellular entry into human cells may be small. This may also be the case for the other RhGB01-like sarbecoviruses sampled previously[11]. Strikingly, of the two RhGB01-like sarbecoviruses we investigated here, one was capable of infecting hACE2-overexpressing cells and the other not, despite 98% spike sequence similarity and identical SARS-CoV-2 residues. This further indicates that minor adaptations in the spike protein may significantly affect binding affinity with host receptors, and hence the zoonotic potential of different viral lineages.

We also identified a R-A-K-Q sequence in all European sarbecoviruses that resembles an FCS precursor, but which is absent in all Asian sarbecoviruses considered (Supplementary Fig. 9a). This supports previous observations that FCSs naturally occur in coronaviruses and have emerged independently at least six times amongst betacoronaviruses[53]. However, even after mutating R-A-K-Q to R-A-K-R, we could not detect any cleavage during expression of RhGB07 spike by human proteases (Supplementary Fig. 9b). These findings indicate that, in addition to acquiring a functional FCS via substitution, European sarbecoviruses would likely have to acquire an extended loop structure (like in SARS-CoV-2) via insertion for efficient spike cleavage.

We found a high prevalence of genetic recombination amongst sarbecoviruses, particularly in the spike gene (Supplementary Fig. 10), which may facilitate viral adaptations to overcome the genetic barrier for a zoonotic jump. This observation is corroborated by other studies that have also suggested an enrichment of recombination signals in or surrounding the sarbecovirus spike gene[54,55]. Co-infections and subsequent recombination of RhGB01-like sarbecoviruses with other coronaviruses that already effectively use hACE2 may therefore facilitate zoonotic transmission. As such, the possibility of a future host jump into humans cannot be ruled out, even if the risk is small. This reiterates the need for individuals that are in frequent contact with bats, such as bat rehabilitators, to adhere to current biosafety practices to reduce their exposure to bat coronaviruses and likewise to reduce the risk of the exposure of bats to human-borne coronaviruses[56], such as SARS-CoV-2 or the endemic HCoVs. Fortunately, in the UK, the risk of zoonotic exposure is minimised for most people through a lack of direct contact (roosting spaces are often well away from human inhabitants) along with the provision of science-based information to roost owners by organisations, such as the Bat Conservation Trust (https://www.bats.org.uk).

Our in vitro assays indicate that RhGB01-like sarbecoviruses, including RfGB02 that was directly sampled from this species, do not use *R. ferrumequinum* ACE2 as their primary receptor, which is in line with other studies of bat coronaviruses[43,57]. Importantly, this raises the question as to what evolutionary mechanisms drive the acquisition of the ability to use hACE2 in bat sarbecoviruses. Given previous associations between pathogen host breadth and their capacity to emerge as zoonotic diseases[32,33], we speculate that multi-host viruses tend to have "generalist" cell entry receptors that possess a low genetic barrier to the evolution of zoonotic transmission. More extensive surveillance

of the viral sharing dynamics in mammalian hosts, including bats, may provide key insights into the molecular and ecological determinants of zoonotic events. Such studies can leverage both species occurrence data and niche modelling to prioritise regions where a high number of species are likely to be found combined with an understanding of species ecology for quantification of risk.

The initial spread of SARS-CoV-2 in China, its likely evolutionary origin in *Rhinolophus* bats[54], and the subsequent identification of other bat-borne sarbecoviruses in Southeast Asia[8,10], has focused attentions on the zoonotic risk of coronaviruses in those geographical regions. However, our findings highlight the zoonotic risk of sarbecoviruses may extend beyond Asia, stressing the importance of more extensive surveillance globally.

Finally, whilst it is imperative to quantify the risk of zoonotic events from bats and design approaches to mitigate this risk more effectively, bats fulfil important roles in ecosystems globally, including services such as arthropod pest suppression, pollination and seed dispersal[58]. Some bat species have rapidly declining populations—for example, one third of the most threatened mammalian species in the UK are bats[59,60]. Recent studies have shown that human-associated stressors such as habitat loss and changes in land use can be important drivers of zoonotic spillover from wildlife[61,62], and that bat culls are ineffective in minimising cross-species transmission[63]. As such, it is vitally important that an integrated ecological conservation approach is taken that includes maintaining legal protection, rather than destruction of wildlife and its habitat, in future efforts to mitigate zoonotic risk.

## Methods
### Sample collection
Sampling kits were sent out to various bat rehabilitators in the UK, as described previously[56], for the collection of faeces from bats. These faecal samples (0.02–1 g) were immediately stored in 5 ml of RNAlater solution to prevent degradation of RNA. The geographical locations and collection dates for all samples are provided in Supplementary Data 3. Registered bat rehabilitators have received relevant training to aide in this role, including bat health assessment and identification, public engagement, risk assessment, and legal requirements, approved by experienced trainers. Unlike other activities which involve handling bats, a license is not required for care and rehabilitation purposes in the UK, except where bats are to be kept in captivity for six months or more. Therefore, for this study, faecal samples were collected by bat rehabilitators without the need for legal or ethical approval, in particular because all sampling was non-invasive and did not require the handling of bats.

### Murine hepatitis virus (MHV) spike-in control culture
MHV (GenBank AY700211.1) was propagated in an NCTC 1469 clone derivative (ATCC CCL-9.1) cell line in high glucose DMEM and 10% horse serum. Both the MHV and NCTC cell line were acquired from the American Type culture Collection (ATCC, Manassas, Virginia, USA). Cell culture supernatant was isolated for later RNA extraction.

### RNA extraction
RNA was extracted from faecal samples using the QIAamp Viral RNA Mini Kit (Qiagen) following the protocol for extracting RNA from stool samples. We used up to 0.5 g of faeces, which was vortexed in 2 ml of 0.9% NaCl solution, at 6000 rpm for 2 min. The supernatant was filtered using a 0.2 μm syringe filter, 280 μl of which was used for RNA extraction. For the MHV spike-in control, we used 140 μl of culture supernatant for RNA extraction. Total RNA was eluted in 80 μl of AVE buffer and stored at −80 °C. RNA was quantified using Qubit 2.0 fluorometer (Invitrogen). All faecal extractions were spiked with 20 μl of MHV RNA prior to library preparation to act as a sequencing quality control.

### Coronavirus database
To create a database representing the extant global genomic diversity of coronaviruses, we downloaded all complete *Coronaviridae* (taxid:11118) genomes from NCBI Virus, excluding provirus sequences (accessed July 4, 2022). In addition, we downloaded all non-human-associated and non-SARS-CoV-2 betacoronaviruses from GISAID[64] (n = 29). To minimise the overrepresentation of certain viral species, we randomly retained 50 genomes for each of the following species: porcine epidemic diarrhoea virus, avian infectious bronchitis virus, MERS-CoV, SARS-CoV and SARS-CoV-2 sequences. This yielded a final dataset comprising 2118 genomes.

### Metagenomic sequencing and assembly
All samples were prepared for sequencing using the NEBNext® Ultra™ Directional RNA Library Prep Kit, with a QIAseq FastSelect rRNA depletion step. Prior to library preparation, we also spiked in MHV RNA (GenBank AY700211.1) as a positive control. Sequencing was carried out using Illumina NovaSeq, paired-end 150 bp. Quality control of reads was performed using bbduk.sh v39.01 from the BBTools Suite (https://sourceforge.net/projects/bbmap/). Briefly, we trimmed adapter sequences and read ends below Q10, and discarded trimmed reads with average quality below Q10. Reads that mapped to the positive control using Bowtie2 v2.4.5[65] were removed prior to all downstream analyses. De novo metagenomic assembly was performed on quality-controlled or raw reads for each sample using coronaSPAdes v3.15.4[66]. Assembled scaffolds were then queried using BLASTn against all 2118 genomes in our coronavirus database to determine their most related reference. Scaffolds that could be aligned using BLASTn to coronaviruses in our database and that were already longer than 28 kb were considered as complete genomes.

In some cases, de novo assembly yielded multiple scaffolds that were shorter than 28 kb but shared the same closest reference. We "stitched" these scaffolds together using the BLASTn alignment coordinates to the closest coronavirus reference and replaced any gaps with Ns. De novo assembly using adapter-trimmed reads, without quality trimming or filtering, produced better results, producing longer and more complete scaffolds, yielding six >28 kb scaffolds (MdGB01, MdGB02, MdGB03, PpiGB01, RfGB01, RfGB02), compared to quality-controlled read assembly which yielded only two (RfGB01, PpiGB01). Further, the two >28 kb scaffolds, RfGB01 and PpiGB01, generated using either adapter-trimmed or quality-controlled assemblies were identical, suggesting that de novo assembly using adapter-trimmed reads were reliable. We hence chose the assemblies generated using adapter-trimmed reads for our downstream analyses. We named the novel complete genomes following the naming convention for the sarbecovirus previously described in a UK bat, RhGB01—species: "Rh" (*R. hipposideros*), region the coronavirus was found in: "GB" (Great Britain) and the frequency of description: "01" (the first described in that species and country).

### Genome annotation and characterisation of novel genes
Assembled genomes were annotated using Prokka v1.14.6[67], and annotated genes were inspected to identify and correct erroneous frameshifts that were present in the raw assemblies to produce the final genomes. For the four novel sarbecovirus (RhGB07, RhGB08, RfGb01, RfGB02) and three of the pedacovirus (MdGB01, MdGB02, MdG03) genomes, we also performed genome alignments to their closest known relative shown in Table 1 to check if erroneous indels were present. The gene annotations were also analysed to determine if these genomes carry any novel genes. We used PSI-BLAST on the online webserver (https://blast.ncbi.nlm.nih.gov/), an iterative search programme that is more sensitive than the conventional protein BLAST[68], to identify distant homologues of annotated genes. We additionally used InterProScan[69,70] to make functional predictions for potentially novel proteins.

## Taxonomic classification of sequencing reads

Taxonomic classification of reads was done using Kraken2 v2.1.2[29] with the "–paired" flag and using the "Viral" database maintained by Ben Langmead (June 7, 2022 release; https://genome-idx.s3.amazonaws.com/kraken/k2_viral_20220607.tar.gz). This database comprises all genomes available on NCBI RefSeq as of June 2022. We then extracted reads assigned to each viral family (Supplementary Fig. 3a) or viral species (Supplementary Fig. 3b). To minimise the effects of potential read misclassifications, we applied abundance thresholds as described previously[71]. Briefly, we considered a taxon to be present if greater than 10 read pairs were assigned and if its relative abundance was greater than 0.005.

## Species niche modelling

Bat occurrence records data were gathered from the online databases NBN Atlas (https://nbnatlas.org/) and GBIF (www.gbif.org). Records from year 2000-present were included, removing replicate records and those with high coordinate uncertainty. The number of occurrence points used for modelling ranged from 32 (*Myotis alcathoe*) to 16,403 (*Pipistrellus pipistrellus*). An initial 17 environmental variables were identified a priori to be important for predicting bat distributions. Nine were climatic variables averaged across 1980–2010 as described in ref. 72, and were reduced to five variables using Variance Inflation Factor (VIF), retaining only those with a VIF < 0.5. These were mean annual air temperature, mean diurnal air temperature range, mean daily mean air temperature of the wettest quarter, precipitation seasonality and mean monthly precipitation amount of the warmest quarter. Four variables were derived from the UKCEH Land Cover Map 2019[73]. After merging similar land-use classes, distance to woodland, distance to grassland, distance to arable and horticulture and distance to urban were measured using Euclidean distance tools in ArcMap version 10.8. Two further distance variables were derived from Ordnance Survey polygons (2019, 2021): distance to the nearest road[74] and distance to the nearest river[75]. Elevation and slope were included to describe the topography of Great Britain, and were taken from the LiDAR Composite Digital Terrain Model data at 10m resolution[76]. All spatial data were subsequently reduced to 1000m resolution and projected to British National Grid.

An ensemble of five supervised binary classifiers was trained to predict the suitability of a land area for each of the 17 UK bat species using the R package sdm[77]: random forest (RF), maximum entropy (MaxEnt), multivariate adaptive regression splines (MARS), boosted regression trees (BRT), and support vector machines (SVM). Classifiers were trained to predict the relative probabilities of species occurrence based on the 13 ecological variables described above, using the occurrence data for each species and an equal number of randomly generated pseudo-absence data points across the study area. Training and evaluation were performed using a five-fold cross-validation protocol, where a random subset comprising 80% of the dataset is used for training and the remaining 20% use for the final evaluation. A final ensemble of all five classifiers that were trained was used to generate the species distribution maps, with the contribution of each individual classifier weighted based on its area under the receiver operating characteristic curve (AUROC) score obtained during training. The resultant species distribution maps indicate habitat suitability as a probability score for each 1-km square grid on the study area, which ranges from 0 (unsuitable habitat) to 1 (suitable habitat). All models across all species performed well, with a median AUROC, sensitivity and specificity of 0.827, 0.854 and 0.78, respectively. The individual species distribution maps and model performance metrics are provided in Supplementary Fig. 11 and serve as a useful resource for future studies that seek to understand the geographical range of UK bat species.

## Phylogenetic analyses

To place the novel sequences within the global diversity of coronaviruses sequenced to date, we computed alignment-free pairwise Mash distances using Mash v2.3[31] with a *k*-mer length of 12, and reconstructed neighbour-joining trees[78] using the nj function from the Ape v5.6.2 package in R (Fig. 1a). This alignment-free phylogenetic reconstruction approach circumvents the challenge of aligning highly diverse sequences at the family level, where high frequency of viral recombination may obscure true evolutionary histories[79] and prevent dataset-wide alignments. In accordance with previous work[80], we rooted the neighbour-joining tree to a monophyletic *Deltacoronavirus* clade comprising all 10 representative *Deltacoronavirus* genomes downloaded from NCBI RefSeq.

From this global phylogeny, we retrieved the pedacovirus (*n* = 106), merbecovirus (*n* = 113) and sarbecovirus genomes (*n* = 534) most proximal to the novel assembled genomes. We then aligned genomes from these subgenera separately using the Augur v14.0.0[81] wrapper for MAFFT v7.490[82]. Genome positions where more than 20% of sequences were assigned gaps were removed from the alignment. We subsequently reconstructed finer-scale maximum-likelihood trees with IQTree v2.1.4-beta under a GTR + G model, using ultrafast bootstrapping (UFBoot)[83] and approximate likelihood-ratio tests (SH-aLRT)[84] with 1000 replicates. All phylogenetic trees were visualised either using FigTree v1.4.4 or ggtree v3.2.1[85].

## Recombination analysis

We selected 218 sarbecovirus genomes from the local sarbecovirus tree (*n* = 534) by retaining only one representative each for SARS-CoV (NC_004718) and SARS-CoV-2 (MW206198). We subsequently aligned these genomes via the same approach described above but masked all positions with >20% of gaps by replacing the positions with Ns, and removed gaps in the alignment relative to the genome used to root the local sarbecovirus tree, NC_025217. This masked alignment was then analysed using RDP v4.101[86]. Gene annotations for NC_025217 were obtained from GenBank and used to annotate predicted recombinant positions.

## Spike protein homology and conservation of contact residues

We extracted the Prokka-annotated spike protein sequences from our novel genomes for further analysis. We peformed multiple sequence alignments of spike proteins from our novel genomes and other sarbecoviruses that have been shown to bind human ACE2[25,42,43] (BANAL-236, MZ937003.2; SARS-CoV-2, NC_045512.2; SARS-CoV, NC_004718.3; Rs4084, KY417144.1; RsSHC014, KC881005.1; WIV1, KF367457.1; Rs7327, KY417151.1; Rs4231, KY417146.1; LYRa11, KF569996.1; Pangolin GD-1, EPI_ISL_410721; Pangolin GX-P2V, EPI_ISL_410542; RhGB01, MW719567.1; Khosta-2, MZ190138.1; BtKY72, KY352407.1) using Mafft v7.490[82]. Subsequently, pairwise amino acid similarity scores, visualisation of the alignments, and annotatation were performed using the spike alignments using UGENE v42.0[87]. The accessions of all genome records used in these analyses are provided in Supplementary Data 4.

## Pseudovirus assays

To further test the capability of the coronaviruses we identified to infect human cells, we synthesised human codon-optimised, Δ19-truncated (or equivalent) spike constructs in pcDNA.3.1. The merbecovirus PaGB01 and pedacovirus PpiGB01 were additionally synthesised with GSG-linker Myc tags for detection of spike incorporation into pseudoparticles. Gene synthesis and codon optimisation was performed by GeneArt (Thermo Fisher). Plasmids for human (*Homo sapiens*; BAB40370.1), least horseshoe bat (*Rhinolophus pusillus*; ADN93477.1), Leschenault's rousette fruit bat (*Rousettus leschenaultia*; BAF50705.1), and little brown bat (*Myotis lucifugus*; XP_023609438.1)

in pDisplay were used as previously described[38]. Additionally, Greater horseshoe bat (*Rhinolophus ferrumequinum*; BAH02663.1) ACE2 was synthesised and cloned into pDISPLAY for this study.

We maintained human embryonic kidney cells (HEK293T; ATCC CRL-11268) and human Hepatocyte carcinoma clone 5 (Huh7.5; C. Rice, Rockefeller University, New York, NY) in complete media (DMEM, 10% FBS, 1% non-essential amino acids (NEAA) and 1% penicillin–streptomycin (P/S)). Human lung cancer cells (Calu-3; ATCC HTB-55) and Human epithelial colorectal adenocarcinoma cells (Caco-2; ATCC HTB-37) were maintained in DMEM, 20% FBS, 1% NEAA and 1% P/S. All cells were kept at 5% $CO_2$, 37 °C. 293T-hACE2 and Huh7.5-TMPRSS2 cells were generated by transducing HEK 293 T or Huh7.5 cells with an ACE2 or TMPRSS2-expressing lentiviral vector, MT126 or MT130[88] and selecting with 2 μg ml$^{-1}$ puromycin or 4 mg ml$^{-1}$ G418; after selection, cells were subsequently maintained with 1 μg ml$^{-1}$ puromycin or 2 mg ml$^{-1}$ G418, respectively.

Lentiviral-based pseudotyped viruses were generated as previously described[47]. Briefly, 100-mm dishes of 293T cells were transfected using lipofectamine 3000 (Invitrogen) with a mixture of 1 μg of the HIV packaging plasmid pCAGGs-GAG-POL, 1.5 μg of the luciferase reporter construct (pCSFLW), and 1 μg of the plasmid encoding the spike or glycoprotein of interest in pcDNA3.1. After 24 h supernatant was discarded and replaced. PV-containing supernatants were collected at 48 and 72 h post-transfection, passed through a 0.45-μm filter, and aliquoted and frozen at −80 °C.

Pseudovirus entry assays were performed as previously described[47]. Briefly, 100 mm dishes of 293T cells were transfected using lipofectamine 3000 (Invitrogen) with 2 μg of the ACE2 encoding plasmid or empty vector. After 24 h, cells were resuspended by scraping and plated into 96-well plates. Cells were overlaid with pseudovirus for 48 h before lysis with reporter lysis buffer (Promega). Caco-2, Calu-3, and 293T-hACE2 cells were seeded into 96-well plates. Cells were overlaid with pseudovirus for 48 h before lysis with cell culture lysis buffer (Promega). We determined luciferase luminescence on a FLUOstar Omega plate reader (BMF Labtech) using the Luciferase Assay System (Promega). For all pseudovirus experiments, the amount of pseudovirus added was standardised by quantifying p24 protein by western blot in a matched concentrated pseudovirus stock.

We assessed expression of transfected receptors using Western blot assays. Cell suspensions were pelleted by centrifugation at $200 \times g$ for 7 min at 4 °C, then supernatant was removed. Cells were resuspended in 150 μl of cold radioimmunoprecipitation assay (RIPA) buffer (Thermo Fisher) and incubated on ice for 30 min. Then, they were spun down at 3750 RPM for 30 min at 4 °C. The protein-containing supernatants were transferred to sterile Eppendorfs and frozen down at −20 °C. Before running a gel, 50 μl of 2-Mercaptoethanol (BME; Sigma) diluted 1:10 in 4× Laemmli Sample Buffer (Bio-Rad, USA) was added to lysates and incubated at 80 °C for 10 min.

To analyse incorporation of spike into the different sarbecovirus pseudoviruses, we concentrated pseudovirus by ultracentrifugation at $100,000 \times g$ for 2 h over a 20% sucrose cushion.

In all experiments, we confirmed the successful expression of host receptors and spike pseudoviruses using Western blot analyses (Supplementary Fig. 7c, d, respectively). For western blotting, membranes were probed with mouse anti-tubulin (diluted 1/5,000; Abcam; ab7291), mouse anti-p24 (diluted 1/2000; Abcam; ab9071), rabbit anti-SARS spike protein (diluted 1/2,000; NOVUS; NB100-56578), rabbit anti-HA tag (diluted 1/2000; Abcam; ab9110), rabbit anti-ACE2 antibody (diluted 1/500; Abcam; ab15348), or rabbit anti-Myc tag (diluted 1/2000; Aabcam; ab9106). Near infra-red secondary antibodies, IRDye® 680RD Goat anti-mouse (diluted 1/10,000; Abcam; ab216776), IRDye® 680RD goat anti-rabbit (diluted 1/10,000; Abcam; ab216777), were subsequently used. Western blots were visualised using an

Odyssey DLx Imaging System (LI-COR Biosciences). The ACE2 constructs used in this work are from distinct bat species and exhibited differing levels of expression or stability, as observed previously[38]. Expression levels of ACE2 do not correlate with the efficiency of cell entry. All raw, uncropped western blot images are provided in Supplementary Fig. 12.

## Alphafold2 (ColabFold) structural analysis

The protein structure model of the RhGB07 RBD was predicted using Alphafold2 as implemented in ColabFold[89]. Default settings were used. The top ranked model was used for all analyses. Structural representations and calculations were done within ChimeraX[90,91]. RMSD values for structural superpositions were calculated using the matchmaker command. Reported values represent the RMSD of all $C_\alpha$ atoms. Buried surface area calculations were performed using the measure buriedarea command.

## Biolayer interferometry (BLI)

The RhGB07 spike trimer was designed to mimic the native trimeric conformation of the protein. It consists of a gene synthesised by Genscript of CHO codon-optimised sequence of RhGB07, residues 1–1191, preceded by a u-phosphatase signal peptide[92], residues 969 and 970 mutated to proline (2 P) to stabilise the prefusion state of the spike trimer, a putative basic site that may be the site of proteolysis (RAKQ, residues 669–672, was mutated to GASQ), a C-terminal T4 foldon fusion domain to stabilise the trimer complex, followed by C-terminal 8x His and 2x Strep tags for affinity purification. This gene was cloned with the pcDNA3.1(+) vector. The trimeric RhGB07 spike protein was expressed as previously reported as for the SARS-CoV-2 spike transiently expressed in suspension-adapted ExpiCHO cells (Thermo Fisher) in ProCHO5 medium (Lonza) at $5 \times 10^6$ cells/mL using PEI MAX (Polysciences) for DNA delivery[93]. At 1 h post-transfection, dimethyl sulfoxide (DMSO; AppliChem) was added to 2% (v/v). Following a 7-day incubation with agitation at 31 °C and 4.5% $CO_2$, the cell culture medium was harvested and clarified using a 0.22 μm filter. The conditioned medium was loaded onto Streptactin XT columns (IBA) washed with PBS and eluted with 50 mM biotin in 150 mM NaCl, 100 mM HEPES 7.5. Eluted protein was then dialysed overnight into PBS. The purity of spike trimers was determined to be >99% pure by SDS-PAGE analysis.

Human (residues 19–615), little brown bat (19–629) and greater horseshoe bat (19–615) ACE2 genes were synthesised by Genscript and cloned in after the human pregnancy-specific glycoprotein 1 signal peptide and is followed by a 3C protease cleavage site, a mouse IgG2a Fc fragment and a 10x His tag (only for the hACE2 construct). Protein production was produced exactly as for the RhGB07 spike. The filtered conditioned media was then subjected to Protein A purification. Eluted protein was dialysed into PBS.

Experiments were performed on a Gator BLI system. The running buffer was 1X PBS. Dimeric mFc-hACE2 and bat ACE2 were diluted to 10 μg/mL and captured with MFc tips (GatorBio). Loaded tips were dipped into twofold serial dilution series (highest concentration 3000 nM) of the RhGB07 spike protein. Curves were processed using the Gator software with a 1:1 fit after background subtraction. Plots were generated in Prism v9.

## Data analysis and visualisation

All data analyses were performed using R v4.1.0 or Python v3.9.12. Visualisations were performed using ggplot v3.3.5[94]. Genomic and protein sequence data were analysed using Biostrings v2.62.0[95]. Supplementary Fig. 5 was made using Adobe Illustrator v27.1.1 and Geneious v11.1.5 (https://www.geneious.com).

## Reporting summary

Further information on research design is available in the Nature Portfolio Reporting Summary linked to this article.

## Data availability

All novel genomes are available in NCBI GenBank under the accessions OQ401247-OQ401251 and OQ401253-OQ401255 (BioProject accession PRJNA929706). The raw sequencing reads generated and analysed in this study have also been uploaded to the SRA under the accessions SRX19257406- SRX19257414. All GenBank and GISAID accessions for the sequences included in the coronavirus database are provided in Supplementary Data 4. Other sequences used are as follows: MHV, AY700211.1; BANAL-236, MZ937003.2; SARS-CoV-2, NC_045512.2; SARS-CoV, NC_004718.3; Rs4084, KY417144.1; RsSHC014, KC881005.1; WIV1, KF367457.1; Rs7327, KY417151.1; Rs4231, KY417146.1; LYRa11, KF569996.1; Pangolin GD-1, EPI_ISL_410721; Pangolin GX-P2V, EPI_ISL_410542; RhGB01, MW719567.1; Khosta-2, MZ190138.1; BtKY72, KY352407.1. The NBN Atlas datasets used for species niche modelling are listed in Supplementary Data 5.

## Code availability

All custom codes used to perform the analyses reported here are hosted on GitHub (https://github.com/cednotsed/bat-CoVs.git)[96].

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

## Acknowledgements

We would like to acknowledge the following bat rehabilitators and conservationists for the collection of bat samples that were crucial for this study: Daniel Whitby, Dorset Bat Group, Claire Andrews, Caitlin Woodfield, Emma Turnbull, Elaine Charlson, Rose-Ann Movsovic, Margaret Grimsey, Hazel Ryan, Gareth Harris, Danielle Linton, Sam Smith, Ross Baker, Lynn Whitfield, Colleen Hope, Josh Sowden, Emily Dickens, Tricia Scott, Jonah Tosney, Eilish Rothney, Dale Irvine, Joe Salkeld, Alice Samuel, Amanda Millar, Sheila Wright, Stewart Rowden, Joy Hall, Jessica Dangerfield, Catherine Wood, Rachael Tarlinton, Ternenge Apaa, and Fiona Mathews. We thank Rachael Tarlinton, Ternenge Apaa, and Fiona Mathews for comments on the manuscript, Scott Jones, Faye Hobbs and Danielle Harris for assisting with sample processing, and Xavier Didelot for advice on genetic recombination analyses. We also thank all who had previously deposited on NCBI GenBank and GISAID the genomes used here (Supplementary Data 4 and 6, respectively), as well as the NERC Omics facility for sequencing. V.S., C.C., E.R., G.W. and T.B. were funded by NERC (UKRI Covid Urgency grant). F.B., L.v.D and D.R. are funded by the European Commission (Horizon 2021-2024, END-VOC Project). Views and opinions expressed are, however, those of the authors only and do not necessarily reflect those of the European Union or the European Health and Digital Executive Agency.

## Author contributions

V.S. and F.B. contributed equally and co-supervised the research. V.S., C.M.C., E.R., G.W. and T.B. wrote the grant application that supported this research. Primary analysis of the project was carried out by J.T., C.C.S.T., D.R., T.P.P., K.Y.M., K.L. and C.H. with contributions from F.B., L.v.D., W.D.P., C.D.L.O. and W.B. D.N. contributed to structural analyses; D.B., N.T., J.L.Q. and K.S. contributed to pseudovirus assays. L.K., L.W. and J.N.M. contributed to fieldwork and sampling. C.C.S.T. wrote the initial draft of the manuscript, with subsequent rounds of editing from V.S., F.B. and L.v.D. All authors provided intellectual contributions to the manuscript.

## Competing interests

The authors declare no competing interests.

## Additional information

[1]UCL Genetics Institute, University College London, Gower St, London WC1E 6BT, UK. [2]The Francis Crick Institute, 1 Midland Rd, London NW1 1AT, UK. [3]Georgina Mace Centre for the Living Planet, Department of Life Sciences, Imperial College London, Silwood Park Campus, Ascot SL5 7PY, UK. [4]Department of Infectious Disease, Imperial College London, St Marys Medical School, Paddington, London W2 1PG, UK. [5]Protein Production and Structure Core Facility (PTPSP), School of Life Sciences, École Polytechnique Fédérale de Lausanne, Rte Cantonale, 1015 Lausanne, Switzerland. [6]Laboratory of Biological Electron Microscopy (LBEM), School of Basic Science, École Polytechnique Fédérale de Lausanne, Rte Cantonale, 1015 Lausanne, Switzerland. [7]Queen's Medical Centre, University of Nottingham, Derby Rd, Lenton, Nottingham NG7 2UH, UK. [8]The Pirbright Institute, Surrey GU24 0NF, UK. [9]Nuffield Department of Medicine, University of Oxford, Oxford OX3 7BN, UK. [10]The Bat Conservation Trust, Studio 15 Cloisters House, Cloisters Business Centre, 8 Battersea Park Road, London SW8 4BG, UK. [11]These authors contributed equally: Cedric C.S. Tan, Jahcub Trew, Thomas P. Peacock. ✉e-mail: v.savolainen@imperial.ac.uk

