## [Peer Review File · Nature Communications]

Genomic screening of 16 UK native bat species through conservationist networks uncovers coronaviruses with zoonotic potentialEditorial Note: This manuscript has been previously reviewed at another journal that is not operating a transparent peer review scheme. This document only contains reviewer comments and rebuttal letters for versions considered at *Nature Communications*.

REVIEWERS' COMMENTS

Reviewer #1 (Remarks to the Author):

The authors have mostly addressed my concerns, but I still would like to request two changes before recommending acceptance:

- A bit clearer articulation that binding to ACE2 is just one component of being able to infect humans, so ACE2 binding is not the same as zoonotic capability. This is sort of mentioned, but needs to be stated more clearly at places where zoonotic potential is mentioned.
- Unfortunately I remain concerned about the data in Figure 2. I strongly think the figure should show the raw titers (currently shown in Supp Fig 8), not the normalized titers. While the basic result is the same in both, the visual impression is overly exaggerated in Figure 2 in a way that does not seem fully justified to me when looking at Supp Figure 8. In addition, in current Figure 2, bars of the same height for different viruses can indicate very different entry efficiencies if the "bald" control differs among viruses (which it does). For this reason, the main figure should just show the actual titers: there is no good reason it should not. Furthermore, looking at Supp Fig 8 I am somewhat concerned about the very efficient entry of SARS-CoV-2 into even cells not expressing ACE2. This seems more than is typically reported in the literature. I am not saying it needs to be re-done, but at least it needs to be shown in the main figure as this high level of entry into empty vector cells affects the normalization used in the current Figure 2.

Reviewer #3 (Remarks to the Author):

The revised manuscript by Tan and colleagues addresses most of my initial comments. I believe this version better helps the reader understand the methodological approaches and the limits of some analyses or claims.

I only have minor comments/questions:

Can the authors confirm that all pseudovirus stocks were normalized (like stated for the RhGB01-like spike line 660) for the experiments shown in Fig 2?

For transparency, the fact that the 293T cells transfected with distinct ACE2 appear (Suppl Fig 13) to express different levels (and possibly with variation of levels at the surface of cells) could be mentioned in the main text or Fig 2 legend.

Line 238: I could not find a figure or citation showing the lower levels of ACE2 expression of the stably transduced HEK293T-hACE2 cells (maybe to add on Suppl Fig 13?)

Line 361: please replace homology by similarity.

Line 481 please precise whether the spike-in was performed prior to library preparation (likely considering the analysis report), or just prior to the sequencing itself.

Line 524 precise if raw or trimmed reads

Line 609 replace isolates by genomes

Supplementary Figure 3: still cannot understand the interest of the graph lines with area under curve coloured, considering the small n and the likely incomparable y axis between species.

Response to reviewers

We would like to, once again, thank the editor and all the reviewers for the time and effort put into assessing our manuscript.

We have now revised our manuscript and followed all reviewers' suggestions. Please find below a point-by-point response to the referees' comments (in blue).

Reference to line numbers below are based on the revised manuscript that includes tracked changes, but with 'Simple Markup' toggled.

Reviewer #1

The authors have mostly addressed my concerns, but I still would like to request two changes before recommending acceptance:

1. A bit clearer articulation that binding to ACE2 is just one component of being able to infect humans, so ACE2 binding is not the same as zoonotic capability. This is sort of mentioned, but needs to be stated more clearly at places where zoonotic potential is mentioned.

RESPONSE: We have now added a much more extensive discussion of zoonotic potential in the introduction, to clarify any potential confusion with zoonotic capability (lines 84-108); the new text reads as follows

'For a zoonotic event to occur, a virus must be able to transmit efficiently between animals and humans and be able to infect and replicate in human cells through interactions with host cellular machinery. Additionally, zoonotic pathogens of most concern are those that can transmit efficiently between humans. As such, the true zoonotic potential (i.e., the likelihood of infecting humans in the future) of a virus can only be determined by assessing all these prerequisite abilities. Therefore, excluding unethical experimental infection or transmission studies involving humans, assessments of zoonotic potential should be done via alternative approaches. These include in silico analyses that determine the degree of sequence and structural homology to other known and closely-related human-infecting viruses. Even more compelling evidence for zoonotic potential can be obtained in vitro, and one of the most direct assessments is to isolate and test the infectivity of novel viruses in human cells. However, this would increase the risk of exposure to these potentially infectious agents, necessitating stringent biosafety precautions. Additionally, isolation of novel viruses via cell culture without prior knowledge of their cell tropism and receptor usage can be challenging.

A lower risk and effective alternative is to measure the binding efficiency of viral entry proteins to host receptors, or to assess efficiency of viral entry into human cell lines via a pseudovirus assay, which expresses only the viral entry protein in a non-infectious reporter system. While observed binding and cellular entry in these low-risk assays do not indicate that a virus can replicate effectively in human cells, infect humans, or transmit efficiently between humans, they provide an indication of which human cell receptors can be exploited by novel viruses during infection, which are one key determinant of viral infection.

However, despite the importance of functional validation, many studies to date fall short of providing in vitro or even in silico assessments of zoonotic risk (Supplementary Data 1).'

We have also edited the lines 224-227 to read:

'We tested whether representatives of the newly identified UK coronaviruses (the sarbecoviruses RhGB07 and RfGB02, the merbecovirus PaGB01, and the pedacovirus PpiGB01) could use human cellular receptors for viral entry as a proxy for assessing their zoonotic potential.'

2. Unfortunately I remain concerned about the data in Figure 2. I strongly think the figure should show the raw titers (currently shown in Supp Fig 8), not the normalized titers. While the basic result is the same in both, the visual impression is overly exaggerated in Figure 2 in a way that does not seem fully justified to me when looking at Supp Figure 8. In addition, in current Figure 2, bars of the same height for different viruses can indicate very different entry efficiencies if the "bald" control differs among viruses (which it does). For this reason, the main figure should just show the actual titers: there is no good reason it should not. Furthermore, looking at Supp Fig 8 I am somewhat concerned about the very efficient entry of SARS-CoV-2 into even cells not expressing ACE2. This seems more than is typically reported in the literature. I am not saying it needs to be re-done, but at least it needs to be shown in the main figure as this high level of entry into empty vector cells affects the normalization used in the current Figure 2.

RESPONSE: We have now modified Figure 2 to show both the unnormalised data on the dropping axis (as requested by referee 1), as well as the normalised data on the rising axis (as asked by referee 3, since the usage efficiency of a receptor can only be determined by comparing the change in cellular entry with or without receptor overexpression; i.e. ACE2/DPP4/APN vector vs. empty vector).

We also agree that the entry of SARS-CoV-2 spike pseudoviruses into empty vector cells is not trivial. This may be expected as previous studies have demonstrated appreciable entry of SARS-CoV-2 into human cells such as HEK 293Ts that express negligible levels of ACE2, likely through near undetectable levels of ACE2 and/or the usage of alternative receptors¹⁻⁴.

Reviewer #3

The revised manuscript by Tan and colleagues addresses most of my initial comments. I believe this version better helps the reader understand the methodological approaches and the limits of some analyses or claims.

I only have minor comments/questions:

1. Can the authors confirm that all pseudovirus stocks were normalized (like stated for the RhGB01-like spike line 660) for the experiments shown in Fig 2?

RESPONSE: Yes, we can confirm that all pseudovirus stocks were normalised for the experiments shown in Fig 2. We now explicitly state this in lines 684-686:

‘For all pseudovirus experiments, the amount of pseudovirus added was standardised by quantifying p24 protein by western blot in a matched concentrated pseudovirus stock.’

2. For transparency, the fact that the 293T cells transfected with distinct ACE2 appear (Suppl Fig 13) to express different levels (and possibly with variation of levels at the surface of cells) could be mentioned in the main text or Fig 2 legend.

RESPONSE: As suggested, we have now mention this in the main text (lines 277-278):

‘These bat ACE2 proteins all expressed robustly, albeit to slightly differing efficiencies (Supplementary Figure 7b), as seen previously.’

We also mention this in the Methods section (lines 710-712):

‘The ACE2 constructs used in this work are from distinct bat species and exhibited differing levels of expression or stability, as observed previously. Expression levels of ACE2 do not correlate with efficiency of cell entry.’

3. Line 238: I could not find a figure or citation showing the lower levels of ACE2 expression of the stably transduced HEK293T-hACE2 cells (maybe to add on Suppl Fig 13?)

RESPONSE: Thank you for pointing this out. We now include a new panel in the supplementary figure showing the relative ACE2 expression levels of non-transfected/transduced, transfected, and stably transduced HEK293T cells (Supplementary Fig. 7a).

4. Line 361: please replace homology by similarity.

RESPONSE: We have made the replacement as suggested (line 378).

5. Line 481 please precise whether the spike-in was performed prior to library preparation (likely considering the analysis report), or just prior to the sequencing itself.

RESPONSE: Thank you for pointing out this ambiguity, we now explicitly state that spike-ins were performed before library preparation (lines 505-506):

'Prior to library preparation, we also spiked in MHV RNA (GenBank AY700211.1) as a positive control.'

6. Line 524 precise if raw or trimmed reads

RESPONSE: Thank you for pointing this out. We now explicitly mention that we used adapter-trimmed reads (lines 527-529):

'We hence chose the assemblies generated using adapter-trimmed reads for our downstream analyses.'

7. Line 609 replace isolates by genomes

RESPONSE: We have made the replacement as suggested (line 634).

8. Supplementary Figure 3: still cannot understand the interest of the graph lines with area under curve coloured, considering the small n and the likely incomparable y axis between species.

We have removed the density plot from this Supplementary Figure as we agree with you that it does not provide any meaningful insights given the small sample sizes per bat species (see new Supplementary Fig. 2).

References

1. Partridge, L. J. *et al.* ACE2-independent interaction of SARS-CoV-2 spike protein with human epithelial cells is inhibited by unfractionated heparin. *Cells* **10**, 1419 (2021).
2. Lim, S., Zhang, M. & Chang, T. L. ACE2-Independent Alternative Receptors for SARS-CoV-2. *Viruses* **14**, 2535 (2022).
3. Shen, X.-R. *et al.* ACE2-independent infection of T lymphocytes by SARS-CoV-2. *Signal Transduct. Target. Ther.* **7**, 83 (2022).
4. Dicken, S. J. *et al.* Characterisation of B. 1.1. 7 and Pangolin coronavirus spike provides insights on the evolutionary trajectory of SARS-CoV-2. *BioRxiv* (2021).